

# RIDER distortions in the CODEX experiments

Alexey Krushelnitsky, Kay Saalwächter

Institute of Physics, Martin-Luther-University Halle-Wittenberg, Halle, 06120, Germany

*Correspondence to*: Alexey Krushelnitsky (krushelnitsky@physik.uni-halle.de)

**Abstract.** CSA and dipolar CODEX experiments enable obtaining abundant quantitative information on the reorientation of the CSA and dipolar tensors on the millisecond-second time scales. At the same time,  proper performance of the experiments and data analysis can often be a challenge since CODEX is prone to some interfering effects that may lead to incorrect interpretation of the experimental results. One of the most important such effects is RIDER (Relaxation Induced

Dipolar Exchange with Recoupling). It appears due to the dipolar interaction of the observed X-nuclei with some other nuclei, which causes an apparent decay in the mixing time dependence of the signal intensity reflecting not molecular motion but spin-flips of the adjacent nuclei. This may hamper obtaining correct values of the parameters of molecular mobility. In this contribution we consider in detail the reasons, why the RIDER distortions remain even under decoupling conditions and propose measures to eliminate them. Namely, we suggest the additional Z-filter between the cross-polarization section and

the CODEX recoupling blocks, which suppresses the interfering anti-phase coherence responsible for the X-H RIDER. The experiments were conducted on  rigid model substances  as well as microcrystalline $^2$H/$^{15}$N-enriched  proteins (GB1 and SH3) with a partial back-exchange of labile protons. Standard CSA and dipolar CODEX experiments reveal a fast decaying component in the mixing time dependence of $^{15}$N nuclei in proteins, which can be interpreted as a slow overall protein rocking motion. However, the RIDER-free experimental setup provides flat mixing time dependencies meaning that the

studied proteins do not undergo global motions on the millisecond time scale.

## 1 Introduction

CODEX (Cenralband Only Detection of EXchange) (deAzevedo et al., 1999; deAzevedo et al., 2000; Luz et al., 2002; Reichert and Krushelnitsky, 2018) is a powerful NMR tool for studying molecular dynamics in millisecond to second time scale under the magic angle spinning (MAS). It is based on the stimulated echo principle; the simplified pulse sequence is

shown in Fig. 1. Depending on the phases of the rf-pulses and receiver, one may record a signal, which is proportional to $\sin(\Phi_1)\cdot\sin(\Phi_2)$ (SIN-component) or $\cos(\Phi_1)\cdot\cos(\Phi_2)$ (COS-component), where $\Phi_1$ and $\Phi_2$ are the phases accumulated by the magnetization vector during the precession under recoupling conditions in the dephasing and rephasing periods, respectively. The sum of the two signals (COS and SIN components) is proportional to $\cos(\Phi_1-\Phi_2)$. The classical CODEX experiment was designed for observing the reorientation of the CSA-tensor: the REDOR-like (Gullion and Schaefer, 1989) train of rotor-





synchronised recoupling π-pulses applied on X-nuclei reintroduce the CSA interaction and thus, the phases $\Phi_1$ and $\Phi_2$ are

determined by the precession under the influence of the CSA interaction during the de(re)phasing periods. Potentially

interfering dipolar interactions with protons are supposed to be averaged out by proton decoupling during the de(re)phasing

periods. However, CODEX can be easily modified for observing motionally modulated dipolar interaction or isotropic

chemical shift (i.e. chemical exchange). This can be achieved by a corresponding modification  of the recoupling pulses

(Krushelnitsky et al., 2013; Reichert and Krushelnitsky, 2018).

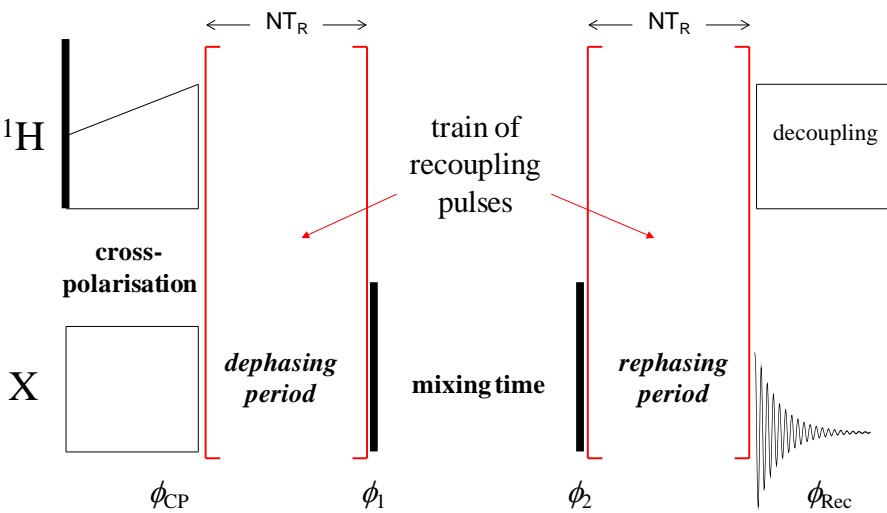

**Figure 1. A simplified scheme of the CODEX pulse sequence. Black vertical bars denote π/2-pulses, $\phi_{CP}$, $\phi_1$, $\phi_2$ and $\phi_{Rec}$ are the**
**phases of the X-channel cross-polarisation pulse, two π/2-pulses and the receiver, respectively. The COS-component is recorded**
**when the phase differences are $\phi_{CP}-\phi_1=\pi/2$ and $\phi_2-\phi_{Rec}=\pi/2$, the SIN-component corresponds to $\phi_{CP}=\phi_1$ and $\phi_2=\phi_{Rec}$.**

In the CODEX experiment, one can measure the signal intensity as a function of both mixing time and the length of the

de(re)phasing periods $NT_R$ ($T_R$ is the MAS period and N is the number of rotor cycles in the de(re)phasing periods), which

provides the information on both time scale and geometry of a molecular motion (Luz et al., 2002). Thus, the CODEX

experiment enables obtaining more abundant quantitative information on molecular dynamics in comparison to standard

NMR relaxation studies. At the same time, CODEX is prone to some interfering effects that may distort the information on

molecular dynamics and that should be taken into account in the data analysis. Two most important effects are the proton-

driven spin diffusion between X-nuclei and RIDER (Relaxation Induced Dipolar Exchange with Recoupling) (Saalwächter

and Schmidt-Rohr, 2000). Spin-diffusion reveals itself as a signal decay in the mixing time dependence, which can be in

some cases erroneously attributed to a molecular motion process. Suppressing the spin-diffusion by proton decoupling

during the mixing time is in principle possible, but rather difficult and not always reliable and effective (Reichert and





Krushelnitsky, 2018). The most robust way of removing the undesirable spin-diffusion effect is a spin dilution, e.g. using natural abundance $^{13}$C or perdeuterated samples.

RIDER also leads to an additional decay in the mixing time dependence. Dipolar interaction of X-nuclei ($S$) with either protons or some other magnetic nuclei present in a sample ($I$), adds two terms of the precessing X-nuclei magnetization - in-phase $S_x\cos(\omega t)$ and anti-phase $2S_yI_z\sin(\omega t)$. The last term is the origin of RIDER, which can be simplistically explained as follows: flips of $I_z$ during the mixing time change the sign of the inter-nuclear dipolar interaction (for 1/2-nuclei) and thus change the sign of the dipolar contribution to the precession frequency. This in turn leads to incomplete rephasing of the $S$-magnetization at the end of the rephasing period and thus to decrease of the signal. Therefore, the characteristic time of the decay in the mixing time dependence due to RIDER is determined by the timescale of $I_z$ flips, that is, $T_1$-relaxation of nuclei $I$. In addition, if the homonuclear dipolar interaction between $I$-spins is significant, spin-diffusion (flip-flops) also contributes to the time scale of RIDER, which can be much shorter than $T_1$ of $I$-spins. The standard way of suppressing RIDER in the CODEX experiments is heteronuclear $I$-$S$ decoupling during the de(re)phasing periods. For some $I$-nuclei with a large quadrupolar moment, e.g. $^{14}$N, decoupling is not effective, and in this case, the only way of removing the undesirable RIDER influence is isotopic editing of a sample.

Our interest in the methodological problems of the CODEX experiments was stimulated by the study of the slow motions in solid proteins. Recently, it was shown by means of $R_{1\rho}$ relaxometry that proteins in a solid state undergo slow overall rocking motion (Ma et al., 2015; Lamley et al., 2015; Kurauskas et al., 2017; Krushelnitsky et al., 2018). The time scale of this motion is tens of microseconds, which is the limit of the time window accessible with $R_{1\rho}$ relaxation experiments. What happens in the (sub)millisecond time scale up to now remained unclear and the CODEX experiments could answer the question, whether the rocking motion extends to longer correlation times or not.

We have thus conducted CSA and dipolar CODEX experiments on $^{15}$N nuclei in $^{15}$N,$^2$H-enriched microcrystalline proteins (SH3 and GB1) with a partial back-exchange of labile protons. These experiments were conducted with a site-specific resolution in 2D $^1$H-$^{15}$N correlation spectrum using indirect proton detection of a signal (Chevelkov et al., 2006; Krushelnitsky et al., 2009). Surprisingly, all peaks in 2D spectra without exception reveal decays in the mixing time dependencies as shown in Fig. 2. The amplitude of the decay and the apparent correlation time of the fast component (around 20 ms) are the same for all residues. This component cannot be due to the proton driven spin diffusion since the time scale of the spin diffusion between $^{15}$N nuclei even in fully protonated proteins is much longer (Krushelnitsky et al., 2006). In the CSA CODEX, this could be the RIDER-effect arising due to the dipolar interaction between $^{15}$N and abundant $^2$H nuclei. On the other hand, in the dipolar CODEX experiment, we observe very similar shapes of the mixing time dependencies with very similar parameters of the fast component. This was observed both for SH3 and GB1 microcrystalline proteins. In the dipolar CODEX experiment, the recoupling π-pulses are applied on the proton channel and thus, the $^{15}$N-$^2$H dipolar interaction should be effectively averaged out by MAS. From this one could conclude, that the observed fast component of the mixing time dependencies is not an artefact and does report on a real overall protein motion in the millisecond time scale.

This would mean that the rocking motion of a protein in a crystal has a very wide correlation times distribution, from micro-
to milliseconds.

However, it turned out that the fast decaying component in the mixing time dependencies is actually a highly non-trivial
artefact. Its nature proved to be more complicated than the simple $^{15}$N-$^{2}$H RIDER-effect. Below we explain the details of the
effects responsible for the appearance of this component and suggest some measures for correct conducting CODEX
experiments and avoiding misinterpretations of CODEX data in proteins as well as other samples having complex isotopic
composition.

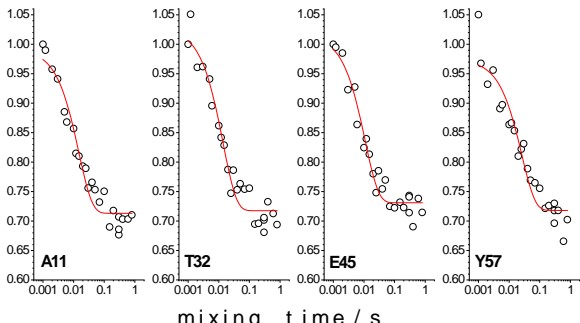

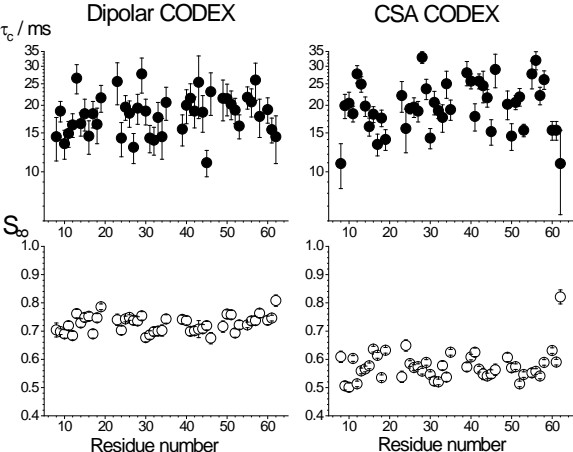

**Figure 2. Results of the residue-resolved dipolar and CSA CODEX experiments in the SH3 protein microcrystalline sample at
ambient temperature, MAS 20 kHz, NT$_R$=2 ms. The mixing times dependences were measured for each resolved peak of the 2D
$^{15}$N-$^{1}$H correlation spectrum. On the top, four examples of the mixing time decays (dipolar CODEX) of backbone $^{15}$N's are shown
for the residues A11, T32, E45 and Y57. Red solid lines are the fits to the simple equation I($\tau_m$)=I(0)·[(1-S$_\infty$)exp(-$\tau_m$/$\tau_c$)+S$_\infty$], where
$\tau_c$ is the apparent correlation time and S$_\infty$ is the decay plateau at long $\tau_m$. At the bottom, $\tau_c$ and S$_\infty$ for dipolar and CSA CODEX
mixing time decays are shown as a function of the residue number.**

## 2 Theory

Here we consider the time evolution of spin coherences in the CSA CODEX experiments using product operator
formalism. It is well known that after $I{\rightarrow}S$ cross-polarization, appear both in-phase $S_x$ and anti-phase -2$S_y I_z$ terms apprear,



see e.g. (Schmidt-Rohr and Spiess, 1994). The anti-phase term is usually neglected since in standard CP/MAS experiments it is suppressed by the heteronuclear proton decoupling during the FID acquisition. In the CSA CODEX, it is supposed to be suppressed by the proton decoupling during the de(re)phasing periods as well. However, in the CODEX pulse sequence this

suppression is much less effective. The train of the X-channel recoupling π-pulses applied during the de(re)phasing periods restores not only CSA, but also dipolar X-[1]H interaction. Hence, the proton decoupling affects not just the residual (after MAS) dipolar interaction, but the restored value of this interaction. For this reason, the small but appreciable dipolar X-[1]H interaction survives during the de(re)phasing periods, which will be demonstrated experimentally below, and we have to take it into account in our analysis. Let us consider the time evolution of the in-phase and anti-phase terms in the CSA CODEX

experiment under the simultaneous influence of the CSA and (not completely suppressed) dipolar interactions during the de(re)phasing periods. The phases acquired during the dephasing period under the influence of the CSA and dipolar interactions we denote as $\Phi_{CSA}$ and $\Phi_D$, respectively. We assume for simplicity that $\Phi_{CSA}$ remains the same for both dephasing and rephasing periods, but $\Phi_D$ can change due to RIDER. Thus, for the rephasing period, the acquired phase will be denoted as $\Phi_D+\Delta\Phi_D$.

In-phase term, dephasing period:

$$S_x \xrightarrow{CSA+DD} S_x \cos(\Phi_{CSA})\cos(\Phi_D) - 2S_xI_z \sin(\Phi_{CSA})\sin(\Phi_D) + S_y \sin(\Phi_{CSA})\cos(\Phi_D) + 2S_yI_z \cos(\Phi_{CSA})\sin(\Phi_D) \quad (1)$$

The first two terms are picked up in the COS-component and two second terms - in the SIN-component. At the end of the rephasing period, we have the COS-component:

$$S_x \cos(\Phi_{CSA})\cos(\Phi_D) - 2S_xI_z \sin(\Phi_{CSA})\sin(\Phi_D)$$

$$\xrightarrow{CSA+DD} S_x(\cos^2(\Phi_{CSA})\cos(\Phi_D)\cos(\Phi_D + \Delta\Phi_D) + \sin^2(\Phi_{CSA})\sin(\Phi_D)\sin(\Phi_D + \Delta\Phi_D)) \quad (2)$$

and the SIN-component:

$$S_y \sin(\Phi_{CSA})\cos(\Phi_D) + 2S_yI_z \cos(\Phi_{CSA})\sin(\Phi_D) \xrightarrow{CSA+DD} S_y(\sin(\Phi_{CSA})\cos(\Phi_{CSA})\cos(\Phi_D)\cos(\Phi_D + \Delta\Phi_D) -$$

$$\sin(\Phi_{CSA})\cos(\Phi_{CSA})\sin(\Phi_D)\sin(\Phi_D + \Delta\Phi_D)) \quad (3)$$

In Eqs. (2) and (3) we left only observable terms that correspond only to COS and SIN components, respectively.

Because of the proton decoupling, the phases $\Phi_D$ and $\Phi_D+\Delta\Phi_D$ are rather small. Thus, we can reasonably assume that

$$\sin(\Phi_D)\sin(\Phi_D + \Delta\Phi_D) \ll \cos(\Phi_D)\cos(\Phi_D + \Delta\Phi_D) \quad (4)$$

$$\text{and} \quad \cos(\Phi_D) = \cos(\Phi_D + \Delta\Phi_D) \text{ for spin I=1/2, since } \Delta\Phi_D \text{ can be either 0 or -2}\Phi_D. \quad (5)$$

which means that for the in-phase term, the effect of the incomplete suppression of the dipolar X-[1]H interaction is almost negligible, it leads only to a small decrease of the signal, proportional to $\cos^2(\Phi_D)$.

Let us now consider the time evolution of the antiphase term. At the end of the dephasing period we have:

$$-2S_yI_z \xrightarrow{CSA+DD} S_x \cos(\Phi_{CSA})\sin(\Phi_D) + 2S_xI_z \sin(\Phi_{CSA})\cos(\Phi_D) + S_y \sin(\Phi_{CSA})\sin(\Phi_D) -$$

$$2S_yI_z \cos(\Phi_{CSA})\cos(\Phi_D) \quad (6)$$


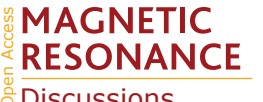

Analogously to Eq. (1), the first two terms in Eq.(5) correspond to the COS-component and the second two terms to the SIN-component. After the rephasing period, the COS-component reads:


$$S_x \cos(\Phi_{CSA}) \sin(\Phi_D) + 2S_x I_z \sin(\Phi_{CSA}) \cos(\Phi_D)$$

$$\xrightarrow{CSA+DD} S_x\{\cos^2(\Phi_{CSA}) \sin(\Phi_D) \cos(\Phi_D + \Delta\Phi_D) - \sin^2(\Phi_{CSA}) \cos(\Phi_D) \sin(\Phi_D + \Delta\Phi_D)\} \qquad (7)$$

and the SIN-component is:

$$S_y \sin(\Phi_{CSA}) \sin(\Phi_D) - 2S_y I_z \cos(\Phi_{CSA}) \cos(\Phi_D) \xrightarrow{CSA+DD} S_y \cos(\Phi_{CSA}) \sin(\Phi_{CSA})\{\sin(\Phi_D) \cos(\Phi_D + \Delta\Phi_D) +$$

$$\cos(\Phi_D) \sin(\Phi_D + \Delta\Phi_D)\}. \qquad (8)$$

Again, in Eqs. (7) and (8) only the observable terms are left that correspond to the COS (Eq. 7) and the SIN (Eq. 8) components. It is seen from these equations that for the anti-phase term, the RIDER effect is not negligible and the inequality analogous to Eq. (4) cannot be written if $\Phi_D$ is small but appreciable.

But how can the RIDER effect arising from the anti-phase term be recognized in the analysis of experimental data? This is relatively simple: one may compare the shapes of the mixing time dependence of the COS and SIN components. If these
curves, namely the ratio $S_\infty/S_0$ ($S_0$ and $S_\infty$ are the signal amplitudes at very short and very long mixing times, respectively), are not similar, than RIDER is relevant. In general, the ratio $S_\infty/S_0$ for the COS and SIN components should be exactly the same, if only molecular motions and/or spin-diffusion are present in a sample. This can be proved as follows. Let us denote the phases acquired during the dephasing and rephasing periods as $\Phi$ and $\Phi+\Delta\Phi$, respectively. At short mixing times, $\Delta\Phi=0$, then the ratio $S_\infty/S_0$ for different cases would be as follows.

Classical CODEX (COS+SIN components):

$$\frac{S_\infty}{S_0} = \langle \cos(\Phi)\cos(\Phi + \Delta\Phi) + \sin(\Phi)\sin(\Phi + \Delta\Phi) \rangle = \langle \cos(\Delta\Phi) \rangle$$

$$(9)$$

COS component:

$$\frac{S_\infty}{S_0} = \frac{\langle \cos(\Phi)\cos(\Phi + \Delta\Phi) \rangle}{\cos^2(\Phi)} = \frac{\langle \cos(\Phi)(\cos(\Phi)\cos(\Delta\Phi) - \sin(\Phi)\sin(\Delta\Phi)) \rangle}{\cos^2(\Phi)} = \langle \cos(\Delta\Phi) \rangle - \frac{\sin(\Phi)}{\cos^2(\Phi)} \langle \sin(\Delta\Phi) \rangle$$

$$, \qquad (10)$$

SIN component:

$$\frac{S_\infty}{S_0} = \frac{\langle \sin(\Phi)\sin(\Phi + \Delta\Phi) \rangle}{\sin^2(\Phi)} = \frac{\langle \sin(\Phi)(\sin(\Phi)\cos(\Delta\Phi) + \cos(\Phi)\sin(\Delta\Phi)) \rangle}{\sin^2(\Phi)} = \langle \cos(\Delta\Phi) \rangle + \frac{\cos(\Phi)}{\sin^2(\Phi)} \langle \sin(\Delta\Phi) \rangle$$

. $\qquad (11)$

Next, we have to recall that $\Delta\Phi_{ij} = -\Delta\Phi_{ji}$ ($i$ and $j$ are the numbers of the exchanging sites) and since it is always assumed that we are dealing with dynamic equilibrium (i.e. the populations of the exchanging sites are constant in time), then obviously $\langle \sin(\Delta\Phi) \rangle = 0$. Thus, in all cases $\frac{S_\infty}{S_0} = \langle \cos(\Delta\Phi) \rangle$, that is the shapes of the COS and SIN components should be the same, although the absolute amplitudes in general case are of course different.

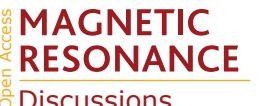

Now, let us estimate the ratio $S_\infty/S_0$ for the COS and SIN components described in Eqs.(7) and (8) taking into account

Eq.(5) and the equation $\langle sin(\Delta\Phi_D)\rangle = \langle sin(\Phi_D + \Delta\Phi_D)\rangle = 0$ (note that this is valid only for I=1/2). COS-component:

$$\frac{S_\infty}{S_0} = \frac{\cos^2(\Phi_{CSA})\sin(\Phi_D)\cos(\Phi_D) - \sin^2(\Phi_{CSA})\cos(\Phi_D)\langle\sin(\Phi_D + \Delta\Phi_D)\rangle}{\sin(\Phi_D)\cos(\Phi_D)(\cos^2(\Phi_{CSA}) - \sin^2(\Phi_{CSA}))} = \frac{\cos^2(\Phi_{CSA})}{\cos^2(\Phi_{CSA}) - \sin^2(\Phi_{CSA})}$$

(12)

SIN-component:

$$\frac{S_\infty}{S_0} = \frac{\sin(\Phi_{CSA})\cos(\Phi_{CSA})(\sin(\Phi_D)\cos(\Phi_D) + \cos(\Phi_D)\langle\sin(\Phi_D + \Delta\Phi_D)\rangle)}{2\sin(\Phi_D)\cos(\Phi_D)\sin(\Phi_{CSA})\cos(\Phi_{CSA})} = \frac{1}{2}$$

165   (13)

Hence, it is clear that the RIDER effect leads to different shapes of the mixing time dependence of the COS and SIN

components. Note that if $\Phi_D$ is not small, the ratio $S_\infty/S_0$ would be different for the COS and SIN components also for the in-

phase term, see Eqs. (2) and (3). From the Eqs. (2), (3), (7) and (8) it can also be deduced that the SIN-component is about

twice more prone to the RIDER-distortions than the COS-component. This follows from the comparison of the amplitudes of

different coherences. The amplitudes of the COS-component of the in-phase and anti-phase terms, see Eqs. (2) and (7), are

proportional to $\cos^2(\Phi_{CSA}) \cdot \cos^2(\Phi_D)$ and $\cos^2(\Phi_{CSA}) \cdot \cos(\Phi_D) \cdot \sin(\Phi_D)$, respectively (here we assume $\Phi_{CSA}$ to be not large).

Hence, the ratio of the anti-phase-term amplitude to the in-phase-term amplitude  is proportional to $\tan(\Phi_D)$ or even smaller

if the second terms in the parentheses in Eqs. (2) and (7) are taken into account. The same ratio for the SIN-component, see

Eqs. (3) and (8), is proportional to $2 \cdot \tan(\Phi_D)$. Thus, the contribution of the anti-phase coherence to the total signal is larger

for the SIN-component.

The analysis presented above is valid only for an isolated *I-S* spin pair. For multinuclear spin systems, the description

would be much more complicated since many types of multiple coherences with a complex network of homo- and hetero-

nuclear dipolar interactions should be taken into account. Quantifying this is outside the frames of our work, still we believe

that on a qualitative level, two most important points remain valid: first, the anti-phase term appearing after the cross-

polarization pulses may cause RIDER distortions of the mixing time dependencies and second, the RIDER effect can be

recognized from the comparison of the shapes of the COS and SIN components. This will be proven experimentally below.

## 3 Materials and methods

### 3.1 Samples

In our work we used four different samples. Model substances: $^{15}$N enriched BOC-Glycine and $^{15}$N enriched Glycine,

which were purchased from Sigma-Aldrich. Proteins: small GB1 and SH3 proteins in a form of microcrystals, $^{15}$N, $^2$H

enriched with a partial back-exchange  of labile protons. The GB1 sample was purchased from Giotto Biotech (Florence,

Italy), the SH3 sample was prepared in Prof. B. Reif's lab (FMP, Berlin). These are the same samples that were used in our



recent work on $R_{1\rho}$ relaxometry (Krushelnitsky et al., 2018). Both protein samples were prepared according to the protocol ensuring 20% of the back-exchange of labile protons. However, we believe that in reality this percentage is somewhat

different: in GB1 it is higher which is indicated by stronger signal and faster proton-driven spin-diffusion between $^{15}$N nuclei (see Figs. 12 and 13 below). The quantitative estimation of this difference is yet rather difficult and uncertain. Since the GB1 sample provides better signal-to-noise ratio, most of the experiments were conducted with this sample.

### 3.2 NMR experiments

The experiments were performed on a Bruker AVANCE II NMR spectrometer (600 MHz) with a 3.2 mm MAS probe. In

the CODEX experiments with the protein samples, the integral intensity of the entire signal was determined without site-selective specification (except for the data shown in Fig. 2). One-dimensional double cross-polarisation ($^1$H→$^{15}$N→$^1$H) proton-detected spectra for SH3 and GB1 proteins were shown in (Krushelnitsky et al., 2018). For the BOC-Gly and Gly samples the direct $^{15}$N or $^{13}$C signal detection was employed, and for the protein samples we used indirect $^1$H signal detection of the $^{15}$N CODEX mixing time dependencies. This was implemented using back cross-polarisation section ($^{15}$N→$^1$H) at the

end of the pulse sequence, according to the approach described earlier (Chevelkov et al., 2006; Krushelnitsky et al., 2009). We have checked - the direct $^{15}$N and indirect $^1$H signal detections in the protein samples provide the same shape of the CODEX mixing time dependencies, in the latter case the signal-to-noise ratio was however better.

To exclude the effect of spin-lattice relaxation during the mixing time, each CODEX mixing time dependence was $T_1$-normalized. For that, for each mixing time dependence two experiments were performed: measuring the mixing time

dependence itself and measuring a $T_1$-relaxation curve within the same time range. After that, the mixing time dependence was divided by $T_1$-relaxation curve. This is a routine procedure described earlier (deAzevedo et al., 1999; deAzevedo et al., 2000; Reichert et al., 2001; Reichert and Krushelnitsky, 2018). Below are shown only the $T_1$-normalized mixing time dependencies for all CODEX experiments in protein samples. For BOC-Gly, the $T_1$-normalization was not performed since $^{15}$N $T_1$ in this sample was extremely long (800-900 s).

The pulse sequences of the CSA and dipolar CODEX are shown in Figs. 3 and 4. For measuring the mixing time dependence, $\tau_m$ was variable and $\tau_r$ was fixed at 1 ms; for measuring $T_1$-relaxation curve, $\tau_m$ was fixed at 1 ms and $\tau_r$ was variable. The phase cycle for both COS and SIN components consists of 64 steps: 2x spin-temperature inversion (ensuring that the signal decays to zero (Torchia, 1978)) for $T_1$-relaxation, 2x spin-temperature inversion for mixing time, 4x CYCLOPS for the $\pi$/2-pulse after mixing time, 4x CYCLOPS for the $\pi$/2-pulse after $\tau_r$ delay (Reichert et al., 2001). Typical

values for $\pi$/2 pulse for $^1$H and $^{15}$N channels were 1.4-1.8 μs and 6.0-6.5 μs, respectively.

The experimental error in estimation the signal amplitude was: less than 1% for BOC-Gly, 1-2% for GB1, 2-4% for SH3 and 5-10% for natural abundance $^{13}$C in Gly. On top of the signal noise, a certain contribution to the experimental error in the mixing time dependences comes from the instability of the MAS controller; that was however significant only for BOC-Gly. The final error of the mixing time dependencies for this sample was around 1-2%. For better visual distinguishing



between the mixing time dependencies shown in Figs. 5, 7 and 8, the adjacent averaging over 5 points filter was applied to the experimental curves in these figures, which significantly decreased the noise spread of the points without the change of the overall shape of the curves.

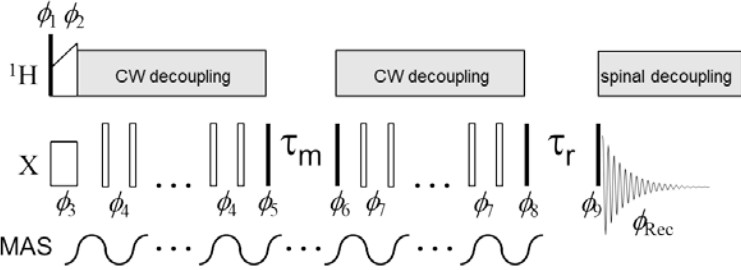

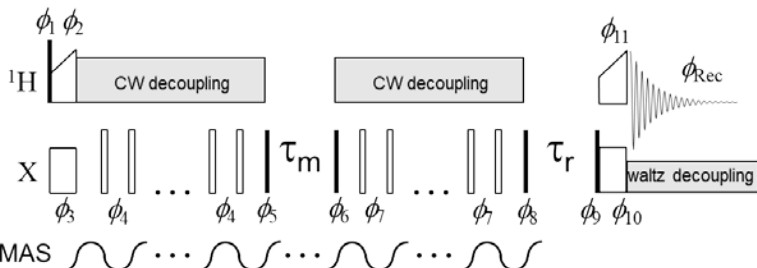

**Figure 3. CSA CODEX pulse sequence for the direct ($^{13}$C or $^{15}$N, top) and indirect ($^1$H, bottom) signal detection. Solid and open bars denote $\pi/2$ and $\pi$ pulses, respectively. The mixing time $\tau_m$ is an integer multiple of the MAS period, which is achieved by MAS rotor triggering before and at the end of the mixing time (see details in Reichert and Krushelnitsky, 2018) . Rotor synchronization during the $\tau_r$-delay is not necessary. Waltz decoupling in the indirect detection sequence aims to suppress only J-coupling between X and $^1$H nuclei, therefore it has low amplitude (few hundred Hz). An additional initial Z-filter and $^2$H-decoupling (see below) are**
**not shown.**

**Phase cycle:**
**$\phi_1$=x; $\phi_2$=y; $\phi_3$=x; $\phi_4$=x; $\phi_5$=(y, -y) (COS component); $\phi_5$=(x, -x) (SIN component);**
**$\phi_6$=(x, x, y, y, -x, -x, -y, -y); $\phi_7$=(y, -y, -x, x, -y, y, x, -x);**
**$\phi_8$=(-x, -x, -y, -y, x, x, y, y, x, x, y, y, -x, -x, -y, -y) (COS component);**
**$\phi_8$=(y, y, -x, -x, -y, -y, x, x, -y, -y, x, x, y, y, -x, -x) (SIN component);**
**$\phi_9$=(x*16, y*16, -x*16, -y*16); $\phi_{10}$=(y*16, -x*16, -y*16, x*16); $\phi_{11}$=(x, x, y, y, -x, -x, -y, -y);**
**$\phi_{Rec}$=((y,-y)*4, (-y,y)*4, (-x,x)*4, (x,-x)*4, (-y,y)*4, (y,-y)*4, (x,-x)*4, (-x,x)*4)) (direct detection);**
**$\phi_{Rec}$=(x,- x, y, -y, -x, x, -y, y, -x, x, -y, y, x, -x, y, -y) (indirect detection).**



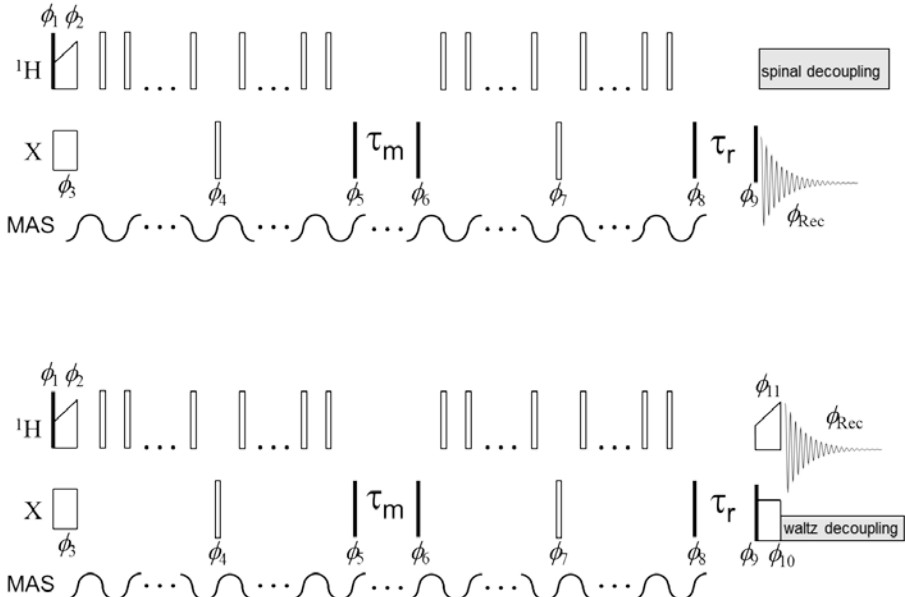


**Figure 4. Dipolar CODEX pulse sequence for the direct (top) and indirect (bottom) signal detection. The denotations are the same as in Fig. 3. $\pi$-pulses applied on X-channel are set in the middle of the de(re)phasing periods, therefore the duration of these periods should be an even multiple of the MAS period. The phase cycle is identical to that shown in Fig. 3. The phases of the $\pi$-pulses applied during the de(re)phasing periods on $^1$H-channel have no critical significance.**

## 4 Results and Discussion

### 4.1 CSA CODEX

#### 4.1.1 Rigid model substances

$^{15}$N-enriched BOC-Glycine is a rigid solid sample in which we do not expect any molecular motion on the millisecond time scale. Thus, the CODEX mixing time decays can be only due to the RIDER effect since the proton-driven spin-diffusion between $^{15}$N nuclei in BOC-Gly is very slow (Krushelnitsky et al., 2006). First, we demonstrate that the anti-phase term discussed above does really cause RIDER distortions in the CSA CODEX mixing time dependence. The anti-phase term appears in the course of cross-polarization; thus, its contribution to the total CODEX signal should depend on the CP contact time. The CSA CODEX mixing time dependencies at various CP times are shown in Fig. 5. These data fully confirm the qualitative theoretical analysis presented above. One may see that the amplitude of the RIDER decay depends on the CP time, that COS and SIN components of the mixing time dependencies are different and that the SIN component is more prone to the RIDER distortions than the COS component. It is interesting to mention that the mixing time dependencies shown in Fig. 5 reveal the decays on two different time scales: a few milliseconds and a few hundred milliseconds. Such a two-component shape of the decays reflects two different mechanisms that cause proton spin flips mentioned in the Introduction above: spin diffusion (flip-flops) and spin-lattice relaxation.






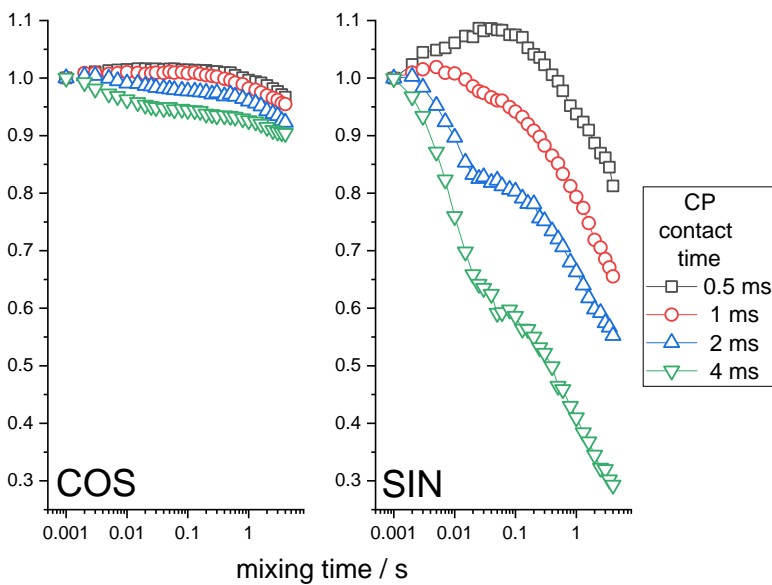

**Figure 5. COS and SIN components of the $^{15}$N CSA CODEX mixing time dependence measured at various cross-polarisation**
**contact times in BOC-Gly. The initial amplitudes of the $\tau_m$-dependencies were normalized to the same value. MAS 20 kHz, NT$_R$ 2**
**ms, 108 kHz $^1$H CW-decoupling during the de(re)phasing periods.**

    If the heteronuclear proton decoupling during the de(re)phasing periods was effective enough, than the RIDER effect
caused by the anti-phase coherence could have been of course avoided. However, this is not always possible for practical
reasons because of the hardware limitations for the power of the long proton pulses. We tried to optimize the proton
decoupling by the maximum signal at short mixing times. Different decoupling schemes were checked (TPPI, WALZ,
SPINAL) at maximum proton power around 100-130 kHz, but none of them provided much better efficiency than simple
CW decoupling (which is not the case for $^1$H-decoupling during FID, where CW is not the best choice). Therefore, in the
experiments shown here we used CW $^1$H decoupling during the de(re)phasing periods in the CSA CODEX measurements.
We do not claim that CW decoupling is the best option for this purpose. It is quite possible that some other decoupling
schemes specifically designed for the case of the recoupling X-pulses can perform better. However, even having such a
decoupling scheme at hand, one should carefully optimize it for different MAS rates and $^1$H field strengths. We suggest here
another, more simple and robust way of suppressing the undesired RIDER effect.

    The anti-phase term can simply be suppressed by an additional Z-filter between the CP pulses and the dephasing period,
as illustrated in Fig. 6. The delay of this Z-filter should be short compared to $^{15}$N $T_1$ and long compared to $T_2$. Thus, after
such Z-filter one would have only in-phase component. Fig. 7 shows the mixing time dependencies of the COS and SIN



components at different delays of the Z-filter. It is clearly seen that the Z-filter fully removes the contribution of the anti-phase coherence.

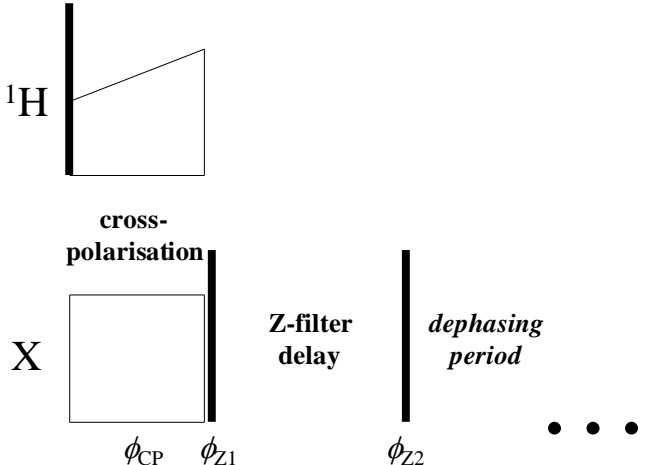

**Figure 6. Initial part of the CODEX pulse sequence (Figs. 1 and 3) with the additional Z-filter between the cross-polarization section and the dephasing period. $\phi_{CP}$-$\phi_{Z1}$=±π/2, $\phi_{Z2}$=-$\phi_{Z1}$.**

Still it is seen that even at long delays of the Z-filter, the mixing time dependencies are not completely flat, as they should be. The observed distortions are obviously the RIDER effect of the in-phase coherence. If the second terms in the

parentheses in Eqs. (2) and (3) are not negligibly small, then the RIDER is present also in the in-phase coherence and the Z-filter cannot remove it. Fig. 8 presents the COS and SIN components of the mixing times dependencies at different durations of the de(re)phasing periods measured with the additional Z-filter. It is clearly seen: the longer $NT_R$, the larger the RIDER distortions. This is reasonable since $\Phi_D$ is proportional to $NT_R$. Note that the COS component is less prone to distortions not only in the case of the "anti-phase", but also of the "in-phase" RIDER. We are not able at the moment to explain the unusual

bell-shaped form of the mixing time dependencies. It is likely that  the network of multi-nuclear dipolar interactions should be taken into account and thus the explanation will not be simple. We also cannot exclude that transient NOE effects may play a certain role.

However, in any case this is the unwanted distortion and regardless of the exact nature of this distortion it should be maximally suppressed in real experiments. For this, the efficiency of the $^1$H-decoupling during the de(re)phasing periods

must be optimized as far as possible. As mentioned above, the standard heteronuclear decoupling schemes used for FID detection do not help much for the case of de(re)phasing periods. This is illustrated by the example of SPINAL sequence, see Fig. 8. Still one may minimize the "in-phase" RIDER effect by keeping $NT_R$ as short as possible and by recording only the COS component of the mixing time dependence. Anyway, the "in-phase" RIDER is much smaller than the "anti-phase" one and in most real experiments it can be safely neglected, as we will see below by the example of the protein samples.





At the end of this section, we demonstrate the application of the additional Z-filter to the natural abundance $^{13}$C CSA CODEX experiment performed on carbonyl carbons in $^{15}$N-enriched glycine ($^{15}$N enrichment is necessary to avoid the $^{13}$C-$^{14}$N RIDER effect). We see the same effect as in the case of $^{15}$N CSA CODEX (Fig. 9).

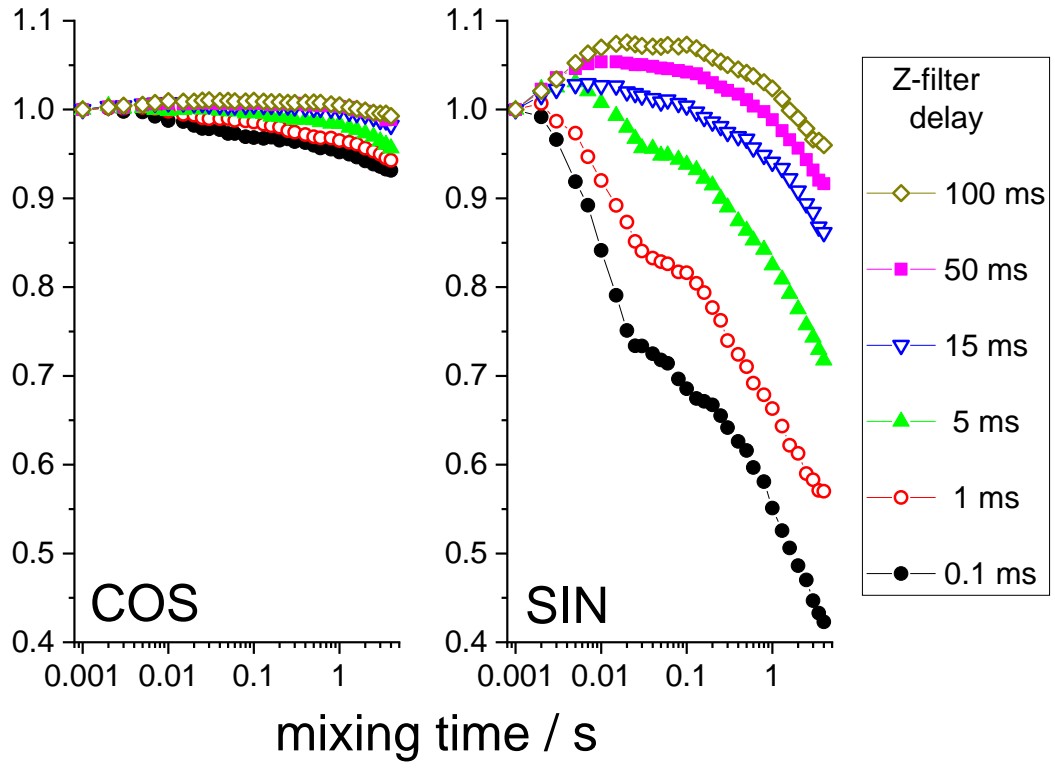

**Figure 7. COS and SIN components of the $^{15}$N CSA CODEX mixing time dependence in BOC-Gly at different Z-filter delays. All the dependencies were normalized to the same initial amplitude. MAS 20 kHz, 108 kHz CW $^1$H decoupling during the de(re)phasing periods, NT$_R$ 2 ms, CP contact time 3 ms.**





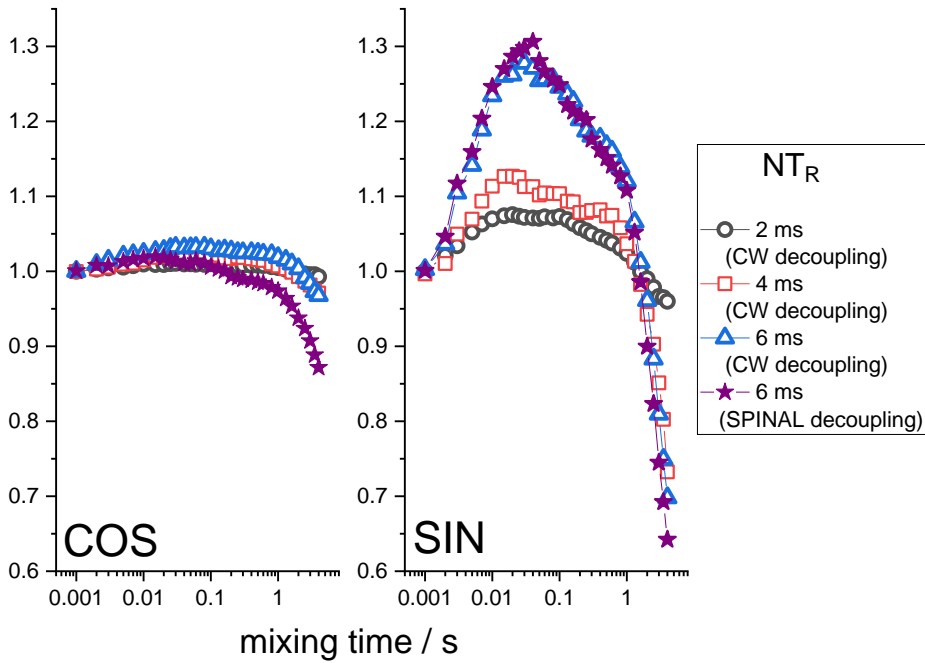

Figure 8. Normalized SIN and COS components of the $^{15}$N CSA CODEX mixing time dependence in BOC-Gly at different NT$_R$. MAS 20 kHz, 108 kHz CW or SPINAL $^1$H decoupling during the de(re)phasing periods, Z-filter delay 100 ms, CP contact time 3 ms.

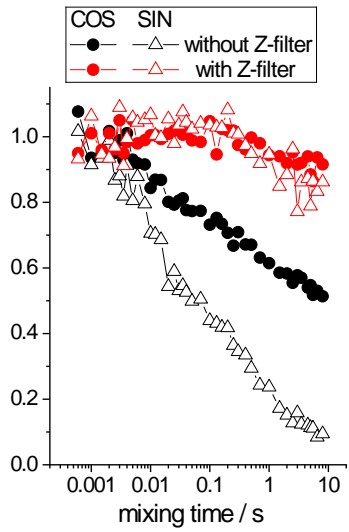

Figure 9. $^{13}$C (carbonyls, nat.abundance) CSA CODEX mixing time dependencies in $^{15}$N-enriched Glycine measured with and without additional Z-filter (Fig. 6). All decays were normalized to the same initial amplitude, the real ratio between the amplitudes of SIN and COS components is 0.7 for both experiments. 80 kHz CW $^1$H decoupling and 35 kHz CW $^{15}$N decoupling during the de(re)phasing periods were applied. MAS 22 kHz, NT$_R$ 2 ms, Z-filter delay 20 ms, CP contact time 3 ms.



### 4.1.2 Protein samples

In the protein samples we have three types of nuclei that we need to take into account - $^{15}N$, $^{1}H$ and $^{2}H$. The direct and inverse $^{1}H$-$^{15}N$ cross-polarisation sections employed in the CODEX pulse sequence ensure that in the experiment we observe only those nitrogens that have protons attached, and all $^{15}N$'s coupled to $^{2}H$ in the protein backbone remain invisible. Still, the interactions between protonated $^{15}N$'s and many remote $^{2}H$'s can be sufficient to induce RIDER-type distortions in the CODEX experiment. To demonstrate the hierarchy of the inter-nuclear interactions in our samples, we measured $^{15}N$ Hahn-
echo decays (Fig. 10) and the initial signal $S_0$ (the signal at short mixing time) in the CSA and dipolar CODEX experiments as a function of $NT_R$ (Fig. 11) for various combinations of $^{1}H$ and $^{2}H$ decoupling schemes. Note that the $S_0$ vs $NT_R$ dependence is in fact the analogue of the Hahn-echo experiment, the only difference is that either CSA or dipolar interaction is reintroduced by means of recoupling pulses during the transverse relaxation.

The conclusions that can be deduced from these data are as follows. First, despite the proton dilution, the $^{15}N$-$^{1}H$ dipolar
line broadening at the MAS frequency 20 kHz remains quite appreciable and strong proton decoupling is necessary to suppress the $^{15}N$-$^{1}H$ dipolar interaction. The comparison of the $S_0$ vs $NT_R$ dependences of dipolar CODEX in fully protonated BOC-Glycine and the deuterated protein shows that the proton dilution reduces of course the inter-proton interaction (flip-flops) and thus, the rate of the $^{15}N$ decay: slower $^{1}H$-flips ensure slower $^{15}N$-$^{1}H$ coupling modulation and hence, better refocusing the signal after the end of the rephasing period. Still the rate of the proton flip-flops in the protein
sample remains in the millisecond range. This is an important point, which will be discussed below. This result corresponds very well to the proton line width estimations made by B. Reif and co-workers (Chevelkov et al., 2006).

Second, it is clearly seen that the 130 kHz CW-decoupling performs much worse in comparison to the SPINAL scheme (Fig. 10). As mentioned above, under the influence of the $^{15}N$ recoupling π-pulses during the de(re)phasing periods, SPINAL does not provide significant advantage in comparison to CW. This confirms our previous statement that the proton
decoupling efficiency under the influence of the X-channel recoupling pulses is much different in comparison to FID detection.





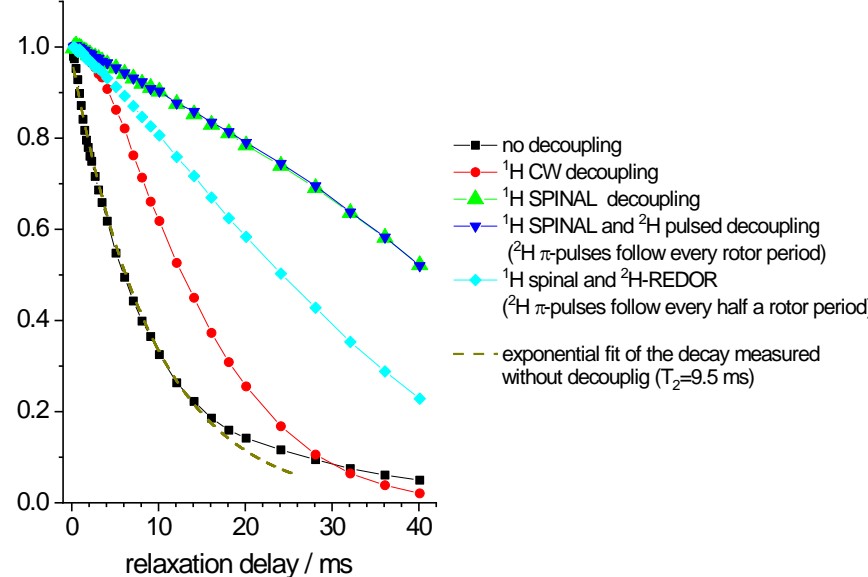

**Figure 10. $^{15}$N Hahn-echo decays measured in GB1 protein sample at different $^{1}$H and $^{2}$H decoupling schemes. MAS 20 kHz, t 13 °C, $^{1}$H decoupling strength (both for CW and SPINAL) 130 kHz, duration of $^{2}$H π-pulses 10.5 μs.**

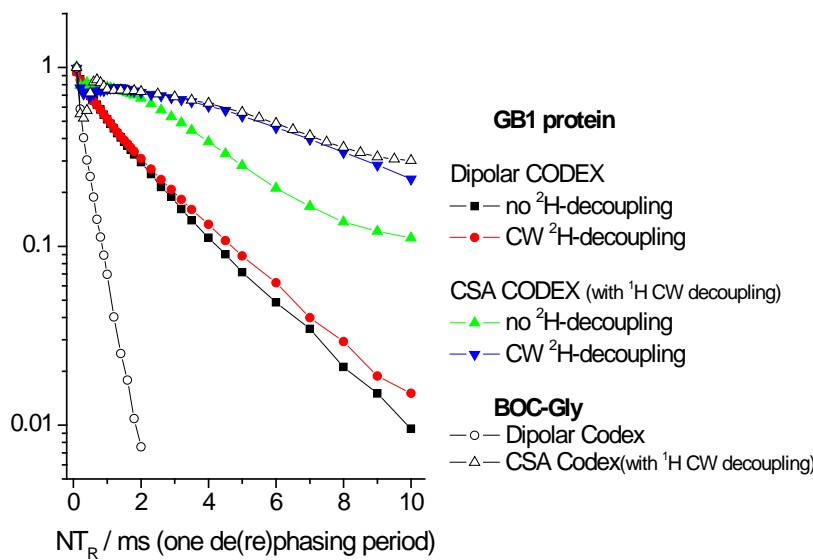

**Figure 11. Signal intensity (COS component) at short mixing time (1 ms) in $^{15}$N CODEX experiments as a function of $NT_R$ in GB1 protein and BOC-Gly samples. MAS 20 kHz, t 13 °C, $^{1}$H and $^{2}$H CW decoupling strengths during the de(re)phasing periods 130 kHz and 45 kHz, respectively.**


Third, the $^{15}N$-$^2H$ interaction is non-negligible. $^2H$ decoupling does not lead to longer the Hahn-echo decays since it is effectively (but not completely, see below) reduced by MAS even without decoupling. However, the reintroduction of the $^{15}N$-$^2H$ dipolar coupling by the REDOR pulse train applied on deuterons appreciably shortens the decays, see Fig. 10. In the CSA CODEX experiment, the $^{15}N$-$^2H$ interaction is initially reintroduced by means of REDOR pulse train applied on $^{15}N$'s. In this case, the $^2H$ decoupling has the effect and makes the decay slower (Fig. 11).

Now the recipe for a methodologically correct CSA CODEX experiment is evident. In deuterated proteins, there are two simultaneous RIDER effects arising from the $^{15}N$-$^1H$ and $^{15}N$-$^2H$ dipolar interactions and one has to take care of both of them. Coincidentally, both RIDERs have similar, although not exactly the same, time constants. The time constant for the proton flip-flops can be estimated directly from the Hahn-decay which gives the value of about 10 ms (Fig. 10). As for the $^2H$ $T_1$, it has a value of 25 ms for aliphatic deuterons in the SH3 protein sample, which was measured by a simple inversion-recovery method. Both these values are quite close to the time constant of the short component of the CODEX mixing time dependences observed in proteins (Fig.2).

The $^{15}N$-$^1H$ and $^{15}N$-$^2H$ RIDER effects can be suppressed by the additional Z-filter between CP and dephasing period (see above) and the rf-decoupling, respectively. Figs. 12 and 13 present the mixing time dependences of the CSA CODEX at various combinations of the $^{15}N$-$^1H$ and $^{15}N$-$^2H$ suppression tools for GB1 and SH3 protein samples. It is seen that the dominant contribution to the short component in the mixing time dependence (Fig.2) comes from the $^{15}N$-$^2H$ RIDER. Still $^2H$-decoupling alone cannot ensure the artefact-free experiment, and only combination of Z-filter and $^2H$-decoupling provides the flat mixing time dependence in the millisecond time scale for both proteins. This demonstrates that both SH3 and GB1 proteins in microcrystalline form do not undergo global motions in the millisecond time scale and the overall rocking motion is limited to the microsecond range only.





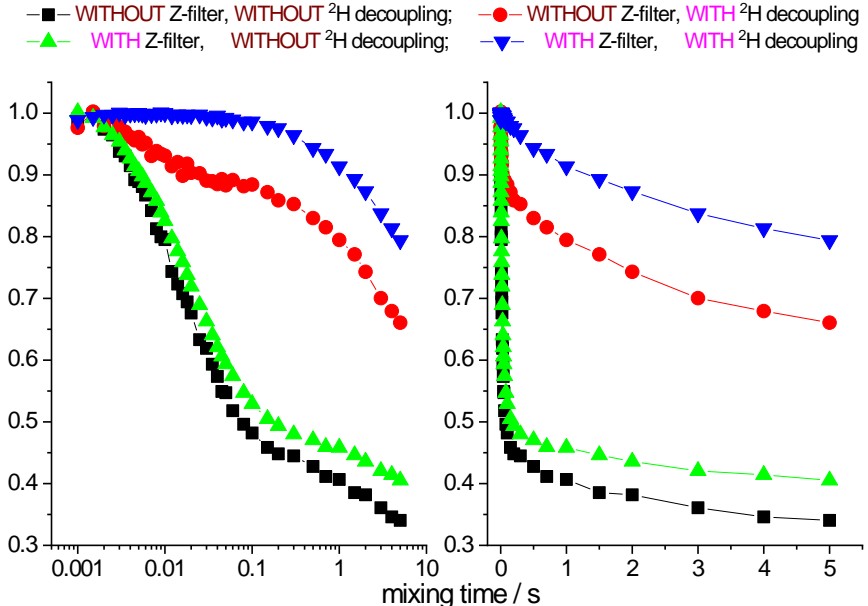

**Figure 12. COS component of the $^{15}$N CSA CODEX mixing time dependence in linear (right) and logarithmic (left) time scale measured in GB1 with/without Z-filter and with/without $^2$H CW decoupling during the de(re)phasing periods. MAS 20 kHz, t 13 ºC, CP contact time 1.5 ms, NT$_R$ 2 ms, $^1$H and $^2$H CW decoupling strengths 130 kHz and 45 kHz, respectively.**

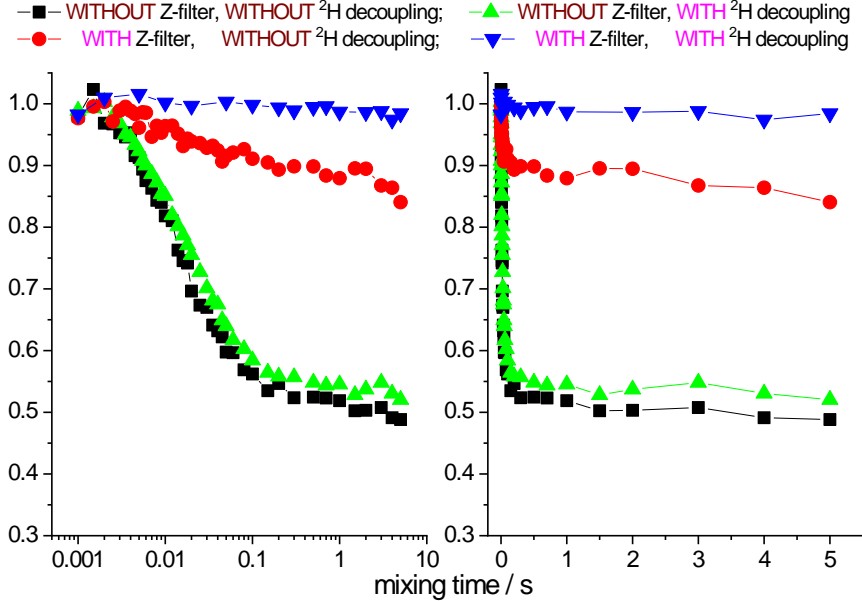

**Figure 13. The same data at the same conditions as shown in Fig. 12 for SH3.**



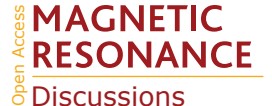

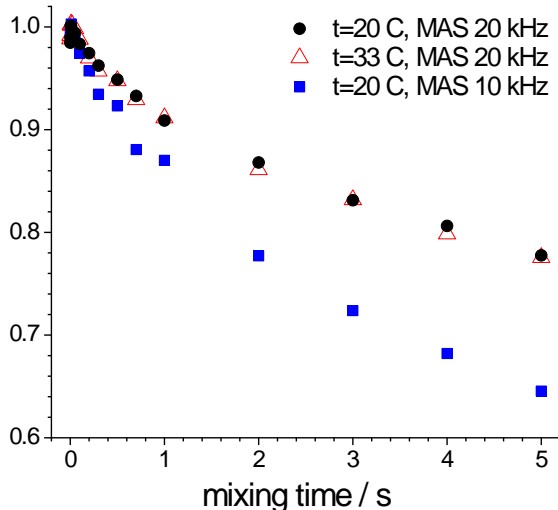

**Figure 14. $^{15}$N CSA CODEX mixing time dependencies measured in GB1 at different MAS rates and temperatures. In all cases Z-filter and $^{2}$H CW decoupling during the de(re)phasing periods were applied (the parameters are the same as mentioned in the caption to Fig. 12). $NT_R$ 2 ms.**

The mixing time dependences in Figs. 12 and 13 also reveal a rather slow decay with a time constant in the second range. This is spin diffusion between $^{15}$N nuclei, which is easy to prove. The spin diffusion rate should not depend on temperature and should depend on the MAS rate (Reichert et al., 2001; Krushelnitsky et al., 2006). We measured the mixing time dependence for the GB1 sample at two temperatures and two MAS rates, see Fig. 14. The results shown in this figure leave no doubts that this is the ordinary proton driven spin-diffusion. The rate of these decays is approximately 3-4 times slower than the spin-diffusion rate in a fully protonated protein (Krushelnitsky et al., 2006) still it is quite appreciable. Spin diffusion rate in SH3 protein is noticeably slower; we believe this is due to the lower proton density in this sample, which is confirmed by a somewhat weaker signal from SH3 compared to GB1.

**4.2 Dipolar CODEX**

The principal problem of the dipolar CODEX is that the $^{15}$N-$^{1}$H interaction cannot be decoupled for obvious reasons and thus the antiphase term responsible for the RIDER effect emerges explicitly during the de(re)phasing periods. To solve this problem, in our first paper on dipolar CODEX (Krushelnitsky et al., 2009) we suggested to measure only the COS component of the mixing time dependence. The COS component must be RIDER-free, which directly follows from the Eq. (2). In the dipolar CODEX $\Phi_{CSA}=0$, and since $\cos(\Phi_D)=\cos(\Phi_D+\Delta\Phi_D)$ (we repeat again that this is valid only for I=1/2), the COS-component of the dipolar CODEX mixing time dependence should not be affected by the $^{15}$N-$^{1}$H RIDER. However, this is only true under the condition that we did not pay a proper attention to at that time. This condition is: the dipolar interaction must be constant during the de(re)phasing periods, i.e. the time scale of the I-spin flips should be much longer








than $NT_R$. If this is not so, then $\cos(\Phi_D) \neq \cos(\Phi_D + \Delta\Phi_D)$ since $\Phi_D$ and $\Phi_D + \Delta\Phi_D$ are randomly modulated by I-spin flips within the de(re)phasing periods. Thus, the COS component at this condition is not RIDER-free anymore.

The comparability of $NT_R$ and the time scale of proton spins filps is exactly our case. We have estimated above the characteristic time of the protons flip-flops, which is about 10 ms (Fig. 10). The duration of the de(re)phasing periods in the
CODEX experiments is usually from few hundred microseconds to several milliseconds. This is shorter than 10 ms but still comparable, which is enough for the RIDER effect. From this we pessimistically conclude that the X-H dipolar CODEX experiment even in proton-diluted samples like deuterated proteins with a partial back-exchange of labile protons is not suitable for studying slow molecular dynamics - there will always be RIDER distortions. This experiment, however, can be implemented using other nuclei pairs, e.g. $^{13}C$-$^{15}N$ (McDermott and Li, 2009), ensuring that the flip-flop time is much longer
than the duration of the de(re)phasing periods.

The last point that deserves to be discussed here is the influence of the $^{15}N$-$^{2}H$ interaction on the dipolar CODEX results. At first sight, there should be no influence, since this interaction is not reintroduced in the dipolar CODEX and it should be simply suppressed by MAS. However, this is not the case. Fig. 15 presents the mixing time dependences in GB1 measured at different powers of the CW-$^{2}H$-decoupling during the de(re)phasing periods. The data demonstrate that in spite of MAS, the
$^{15}N$-$^{2}H$ interaction has a small but well visible contribution to the short component, i.e. RIDER effect, of the mixing time dependence. The residual $^{15}N$-$^{2}H$ interaction is rather small since only few kHz of CW decoupling is enough to suppress it completely. So, why MAS does not do its job alone, without the rf-decoupling? The reason is the protein mobility in the microsecond time scale. If the $^{15}N$-$^{2}H$ interaction is modulated by a molecular motion on a time scale of the MAS period (for 20 kHz it is 50 μs), then MAS cannot suppress this interaction completely, which leads to the increased linewidths of the
MAS centerband (Suwelack et al., 1980). As we know, the correlation time of the protein rocking motion is few tens microseconds (Krushelnitsky et al., 2018). On top of that, there can be interaction of protein nitrogens with deuterons of solvent molecules, and these molecules can also reveal a mobility in the microsecond time scale. Thus, the appearance of the residual $^{15}N$-$^{2}H$ interaction after MAS can be reasonably explained.



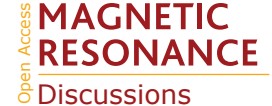

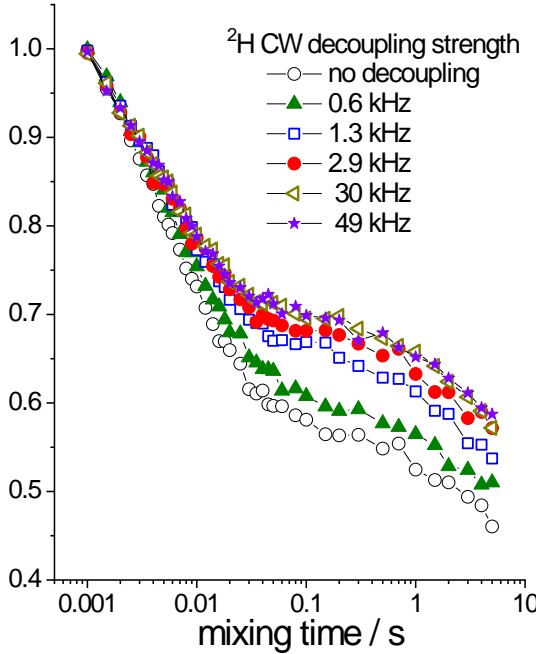


**Figure 15. $^{15}$N dipolar CODEX mixing time dependencies (COS component) measured in GB1 protein sample at various $^{2}$H CW decoupling strengths during the de(re)phasing periods. Z-filter 0.1 s between the cross-polarization section and the dephasing period was applied, MAS 20 kHz, t 13 ºC, NT$_{R}$ 2 ms.**

In summary, we have shown that both $^{15}$N-$^{2}$H and $^{15}$N-$^{1}$H RIDER effects contribute to the short component of the mixing time dependencies of both CSA and dipolar CODEX experiments in the protein samples. However, the dominant contributions in these two experiments are different: in the CSA CODEX the dominant source of the short component is the $^{15}$N-$^{2}$H interaction, and in the dipolar CODEX it is the $^{15}$N-$^{1}$H interaction. As estimated above, the time constants of the two RIDER effects are similar but not the same: $^{2}$H spin-lattice relaxation is somewhat slower than the proton flip-flop rate. Therefore, the apparent decay rate of the short component in the CSA and dipolar CODEX experiments should also be

somewhat different. This is illustrated in Fig. 16, which presents the fast RIDER-components of the CSA and dipolar CODEX experiments after subtraction of the spin-diffusion component and normalization of the decay amplitudes to the same value. The direct comparison of these decays is in a perfect agreement with the findings described above. Interesting to note that in SH3, the difference of the apparent correlation times of the short component for the CSA and dipolar CODEX is much smaller, see Fig. 2 ($\tau_c$ as a function of the residue number). This can also be reasonably explained by the different

proton density in the GB1 and SH3 samples: the lesser the proton density, the slower the flip-flop rate and thus, the smaller the difference between the rates of proton spin diffusion and deuteron spin-lattice relaxation.



**MAGNETIC RESONANCE** Discussions

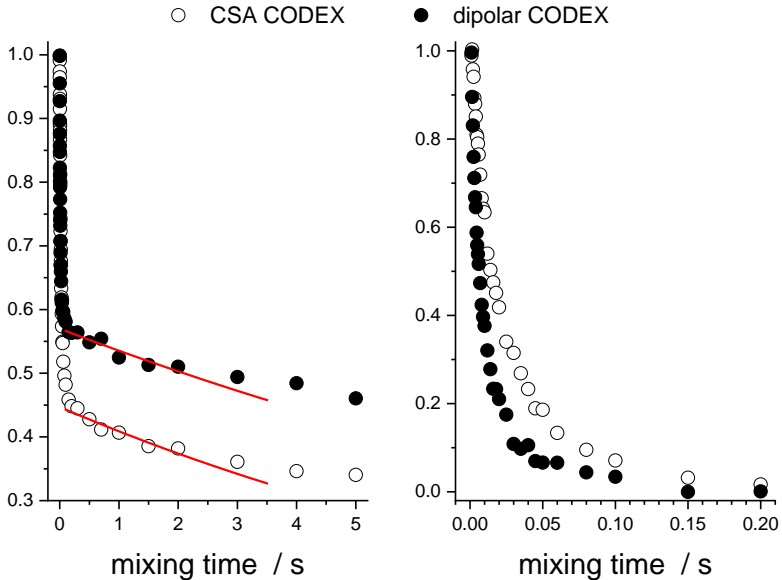

**Figure 16. Direct comparison of the CSA and dipolar CODEX data in GB1 protein sample. Left: the mixing times dependencies taken from Fig. 12 (CSA CODEX, without Z-filter and without $^2$H decoupling) and Fig. 15 (dipolar CODEX, no $^2$H decoupling). Red solid lines - the exponential fits of the initial parts of the spin-diffusion components. Right: Fast initial components of the decays after subtraction the spin-diffusion components and normalization to the same initial amplitude.**

**Conclusions.**

1) The comparison of the shapes of SIN and COS components of the mixing time dependences is a simple and robust way of detecting the presence/absence of the RIDER effect in the CODEX experiments. The COS component is less prone to the RIDER distortion (appearance of the short component) and for minimising this distortion, it is advisable to record and to analyse in experiments only the COS component.

2) Proton decoupling under the influence of the recoupling $\pi$-pulses applied on X-channel is not as effective as in the case of normal X-nuclei FID detection. Thus, the suppression of the antiphase coherence emerging after the cross-polarisation section can be incomplete in CSA CODEX. This may lead to the RIDER distortion in mixing time dependences. This problem can be effectively resolved by inserting additional Z-filter between the cross-polarization section and the dephasing period.

3) In $^{15}$N CODEX experiments in deuterated proteins with a partial back-exchange of labile protons one has to consider two different RIDER effects arising from $^{15}$N-$^1$H and $^{15}$N-$^2$H dipolar interactions. CSA and dipolar CODEX are affected predominantly by $^{15}$N-$^2$H RIDER and $^{15}$N-$^1$H RIDER, respectively. A combination of Z-filter and $^2$H-decoupling during the de(re)phasing periods enables suppression of both effects in the CSA CODEX, however for the dipolar CODEX this is not possible.

4) GB1 and SH3 proteins in their microcrystalline form do not reveal global motion in the millisecond time scale.



**Author contributions**

AK designed the project, conducted the experiments and the data analysis, wrote the paper; KS took part in planning the experiments and discussing the results, and edited the paper.

**Competing interests**

The authors declare no conflict of interests.

**Financial support**

This work was supported by the DFG grant KR 3033/1-1.

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
