# Peer review of "RIDER distortions in the CODEX experiments"

_Magnetic Resonance, 2020_

## Referee Comment (RC1) · Anonymous Referee #1 · 2 Oct 2020

The manuscript by Krushelnitzky and Saalwachter contains a quantitive analysis of the RIDER effect and its influences on the analysis of CODEX dephasing data. CODEX (= Centralband Only Detection of EXchange) allows to study millisecond-sec dynamics under MAS, and involves encoding and decoding of anisotropic interactions (CSA/dipole) by a train of 180 pulses on the X-nucleus. RIDER (= Relaxation Induced Dipolar Exchange with Recoupling) is a consequence of dipolar coupling between X and I nuclei during the de- and rephasing periods and results in Sy.Iz magnetization which is not reconverted in the course of the experiment. An artifact-free version of CODEX is of great interest to expand the range of correlation times to characterize slow motion (rocking motion) in solid proteins.

The authors state that RIDER might be an issue in the CSA CODEX due to 15N-2H

dipolar interactions. This is not clear to me. If 180 pulses on the X channel are applied every half rotor period, this would refocus the evolution of any anisotropic interaction. Please clarify.

Proton driven spin diffusion among X-nuclei is another source that can potentially affect the CODEX dephasing curves. The authors argue that this effect can be neglected, since the buildup of magnetization between X-nuclei takes typically several seconds, and is suppressed by decoupling during the mixing period. In addition, a Z-filter element is suggested to suppress interfering anti-phase coherences.

On the other hand, it is know that CSA facilitates spin diffusion processes (see e.g. Fry EA, Sengupta S, Phan VC, Kuang S, Zilm KW (2011) CSA-Enabled Spin Diffusion Leads to MAS Rate-Dependent T-1's at High Field. J. Am. Chem. Soc. 133: 1156-1158). The CODEX scheme recouples anisotropic interactions. Please comment whether re-coupling of anisotropic interactions enhances spin diffusion.

The authors did not consider ABMS (Anisotropic Bulk Magnetic Susceptibility) as a potential source of artifacts (Vanderhart DL, Earl WL, Garroway AN (1981) Resolution in C-13 NMR of organic-solids using high-power proton decoupling and magic-angle sample spinning. J Magn Reson 44:361–401). Depending on the applied CPMG field, a dephasing effect is observed. Is there a chance that ABMS acts similar as CSA, and enhances spin diffusion effects ? Please comment. This could be tested e.g. by varying the CPMG field in the de- and rephasing CODEX elements.

Minor: Please indicate explicitly the spacing of the 180 pulses in the CODEX de- and rephasing periods.

---

## Referee Comment (RC2) · Anonymous Referee #2 · 7 Oct 2020

The main focus of this article is to trace the origins of the unexpected decay seen in the CODEX dephasing profiles due to the RIDER effect, even when no motion is present in the sample. The authors conclusively show that even when utmost care is taken, the RIDER effect can dominate the decay during the CSA-CODEX mixing time, especially if the antiphase component is not suppressed. The authors clearly demonstrate the features in their experimental datasets that can be used to verify whether the RIDER effect is in play (the difference in the Sinf/S0 ratio of the in-phase and anti-phase component). They further go on to show that the fast-decaying component in GB1 and SH3 is clearly due to the RIDER rather than motion induced CODEX dephasing. Additionally, the RIDER effect in deutarated samples due to 2H-15N dipolar coupling is characterised as well. They go on to suggest a (partial) solution to this problem: the

application of a Z-filter to remove the antiphase component just after CP.

Major comments

The authors have proposed the addition of a Z-filter just after the CP to remove this effect. However, as seen from Fig 7. this does not seem to work at all for the antiphase term in CSA CODEX (which is where one should see a prominent effect). Similarly, for carbonyl, the use of Z-filter is not sufficient to remove these artefacts completely (Fig 9). The final conclusion also seems to be to use only the in-phase component which is less susceptible to this effect, and I would contend that this is the more important of the two. In this light, the abstract, and several statements in the main manuscript appear to me to be misleading when they suggest that the Z-filter takes care of this distortions. Wouldn't something like "Z-mixing and the use of the in-phase component alone" be more appropriate (similar to the final conclusions)

Except in the Fig 12 caption, it is not made clear that only the in-phase component is used for the CODEX data on proteins. I recommend that this be explicitly made clear in the main text.

No pulse is used on 1H during the Z-filter. Will CW decoupling during this period help?

It is unclear what the mechanism for RIDER distortions in the in-phase signal for the CSA-CODEX experiment are (Fig 7, COS component). Is this attributable to imperfect decoupling during the CSA recoupling element?

The conclusions of this article are contrary to the authors' 2009 article on the same subject. This is not clearly bought out, I think it will be very important for the NMR community that this be pointed out clearly and discussed clearly in the manuscript, somewhere near Lines 405-415 (where this is partially done).

Minor Comments

The authors have previously stated (Reichert and Krushelnitsky 2018) that the in-phase component cannot be selected out in a CSA-CODEX experiment. Can this be clarified

in light of Fig. 3?

L 215: "...instability of the MAS controller". How much is this instability, and presumably the rotor synchronization of the pulse sequence by monitoring the output of the mas controller should take care of this?

L 215-220: Does the five-point filter mean that each point shown in the figures is a average of five points, 2 before and 2 after the plotted point? Is this used only for visualisation or for fitting as well?

---

## Author Comment (AC1) · 19 Oct 2020

We thank both reviewers for careful reading the manuscript and making their comments. Below we respond to these comments in detail and describe the changes that we will make in the revised version of the paper.

Reviewer 1

Comment: The authors state that RIDER might be an issue in the CSA CODEX due to 15N-2H dipolar interactions. This is not clear to me. If 180 pulses on the X channel are applied every half rotor period, this would refocus the evolution of any anisotropic interaction. Please clarify.

Reply: We believe this comment is not fully correct. The 180-pulses applied every half

a rotor period do not refocus, they rather reintroduce/recouple all relevant anisotropic interactions (CSA and any heteronuclear dipolar coupling) that are otherwise averaged out by MAS. This is the basic principle of the REDOR pulse sequence and CODEX is actually based on REDOR. The reintroduction of the additional dipolar interaction of the observe nucleus with a third nucleus (e.g. 14N or 2H) in the CSA CODEX experiment was described in detail in the original RIDER paper (Saalwächter and Schmidt-Rohr, 2000).

Comment: Proton driven spin diffusion among X-nuclei is another source that can potentially affect the CODEX dephasing curves. The authors argue that this effect can be neglected, since the buildup of magnetization between X-nuclei takes typically several seconds, and is suppressed by decoupling during the mixing period. In addition, a Z-filter element is suggested to suppress interfering anti-phase coherences. On the other hand, it is know that CSA facilitates spin diffusion processes (see e.g. Fry EA, Sengupta S, Phan VC, Kuang S, Zilm KW (2011) CSA-Enabled Spin Diffusion Leads to MAS Rate-Dependent T-1's at High Field. J. Am. Chem. Soc. 133: 1156-1158). The CODEX scheme recouples anisotropic interactions. Please comment whether recoupling of anisotropic interactions enhances spin diffusion.

Reply: The experimentally measured 15N CODEX mixing time dependencies in proteins that reveal a decay due to the spin-diffusion effect (see our previous publication Krushelnitsky et al. 2006 and Fig. 14 of the present manuscript) are subjects to the CSA recoupling mechanism as well. (In the experiment, we cannot measure proton-driven and CSA-driven spin-diffusion rates separately, we can only measure the integral effect.) Thus, whether the CSA contribution is large or small, it does not matter for the final conclusion: the spin-diffusion between 15N nuclei in proteins in any case is too slow to be responsible for the observed decay with a characteristic time of 20-30 ms. On top of that, Fig. 1 of the paper mentioned by the reviewer demonstrates that at MAS frequencies above 15-16 kHz the characteristic time of the CSA-driven spin-diffusion between 15N nuclei in peptides is of the order of a thousand seconds. We

also note that the recoupling pulses in the CODEX pulse sequence do not affect the spin-diffusion rate: spin-diffusion takes place during the mixing time, when no pulses are applied.

Comment: The authors did not consider ABMS (Anisotropic Bulk Magnetic Susceptibility) as a potential source of artifacts (Vanderhart DL, Earl WL, Garroway AN (1981) Resolution in C-13 NMR of organic-solids using high-power proton decoupling and magic-angle sample spinning. J Magn Reson 44:361–401). Depending on the applied CPMG field, a dephasing effect is observed. Is there a chance that ABMS acts similar as CSA, and enhances spin diffusion effects ? Please comment. This could be tested e.g. by varying the CPMG field in the de- and rephasing CODEX elements.

Reply: The ABMS effect is similar but much weaker than CSA and thus is hardly relevant. In the CODEX pulse sequence one cannot vary the CPMG field (the train of the recoupling 180-pulses) since the time spacing between the pulses is determined by the MAS rate and thus fixed. Unlike the CSA tensor, which may change its orientation due to molecular motions, magnetic susceptibility is a time-independent property, hence this effect cannot cause a decay of the CODEX mixing time dependence in principle. And the last: the additional Z-filter and the deuteron decoupling cannot affect ABMS in any way. Thus, if ABMS is somehow responsible for the decay in the mixing time dependencies, then we would not be able to get flat mixing time dependencies we observe in Figs. 12 and 13 (blue points).

Comment: Minor: Please indicate explicitly the spacing of the 180 pulses in the CODEX de- and rephasing periods.

Reply: The spacing is half a rotor period, as always in the REDOR recoupling. In the revised version of the manuscript, we mention this explicitly in the first paragraph of the Introduction section.

Reviewer 2

Comment: The authors have proposed the addition of a Z-filter just after the CP to remove this effect. However, as seen from Fig 7. this does not seem to work at all for the antiphase term in CSA CODEX (which is where one should see a prominent effect). Similarly, for carbonyl, the use of Z-filter is not sufficient to remove these artefacts completely (Fig 9). The final conclusion also seems to be to use only the in-phase component which is less susceptible to this effect, and I would contend that this is the more important of the two. In this light, the abstract, and several statements in the main manuscript appear to me to be misleading when they suggest that the Z-filter takes care of this distortions. Wouldn't something like "Z-mixing and the use of the in-phase component alone" be more appropriate (similar to the final conclusions)

Reply: We suspect that the reviewer mixes up two different issues: 1) in-phase / anti-phase and 2) COS / SIN contributions to the CODEX signal. The wording "Z-mixing and the use of the in-phase component alone" is not fully correct since the use of a Z-filter ("Z-mixing" is not a proper term in the present context) automatically assumes that we measure only in-phase term. Instead, we suggest to use Z-filter and to measure only COS-component of the signal. That is clearly stated in the final conclusions of the paper: in the conclusion no. 1 we advise to record only the COS-component and in the conclusion no. 2 we advise to use a Z-filter for suppressing the anti-phase coherence. We however admit that the COS-component was not mentioned in the abstract, so in the revised version of the paper we state this explicitly in the abstract.

The statement " However, as seen from Fig 7. this does not seem to work at all for the antiphase term in CSA CODEX (which is where one should see a prominent effect)." is not fully clear to us, probably due to mixing in-phase / anti-phase and COS / SIN decompositions of the signal. In the classical CODEX experiment, both COS and SIN components contain both in-phase and anti-phase terms. Four different combinations of these signal decompositions correspond to Eq. 2 (in-phase, COS), Eq. 3 (in-phase, SIN), Eq. 7 (anti-phase, COS) and Eq. 8 (anti-phase, SIN). We think that Fig. 7 evidently demonstrates the effect of Z-filter on the shape of the mixing time dependence

both for SIN and COS components. This confirms the theoretical consideration: Z-filter suppresses the anti-phase terms as the RIDER distortion becomes smaller upon increasing the Z-filter delay. Why this does not seem to work "at all" is unclear to us, we see from the data that it works quite well. To avoid possible misunderstandings, at the end of the section 4.1.1 we added a new paragraph that briefly summarises the results presented above. This paragraph is below:

" In summary, the theoretical and experimental results presented above show that the proton decoupling under the influence of the recoupling pi-pulses in the CSA CODEX is not fully efficient. This leads to the evolution of both in-phase and anti-phase coherences during the de(re)phasing periods under the influence of the residual 15N(13C)-1H dipolar coupling, that is, to the RIDER effect. The dominant contribution to the RIDER distortions of a mixing time dependence arises from the anti-phase term. This contribution can be suppressed by the additional Z-filter between CP and dephasing period. The RIDER distortion of the in-phase term is smaller but still appreciable at long de(re)phasing periods. This interfering effect cannot be suppressed completely, but it can be significantly minimized if only the COS-component of the mixing time dependence is measured and analyzed since this component is less prone to RIDER in comparison to the SIN-component."

As for the mixing time dependencies of carbonyls (Fig. 9), we think that the main reason of the slow decay measured with the Z-filter is spin-diffusion between natural abundance 13C nuclei. We added few sentences describing this in the text of the paper that we cite below:

" The dependencies measured with the Z-filter (red points in Fig. 9) are not completely flat, however, this is hardly due to the "in-phase" RIDER since the shapes of the SIN and COS components are very similar (in the case of RIDER they should be different) and the time constant of the decay (about 50-60 s) is obviously too long compared to the proton T1 (few seconds). We suspect this decay is a manifestation of the proton-driven spin diffusion between natural abundance 13C nuclei. Its time constant is roughly of the same order of magnitude as spin diffusion times between natural abundance 13C nuclei measured in other organic solids, see e.g. Reichert et al., 1998. Spin diffusion however has no direct relevance to the topic of this work and we did not analyze this in detail."

Comment: Except in the Fig 12 caption, it is not made clear that only the in-phase component is used for the CODEX data on proteins. I recommend that this be explicitly made clear in the main text.

Reply: In all figures it was clearly stated whether the Z-filter was used or not. As mentioned above, the use of the Z-filter automatically assumes that only the in-phase coherence is observed, that was already explicitly mentioned in the text of the paper (lines 279-280 of the original submission). To avoid possible misunderstanding, we added an additional clarification on this issue in the section 4.1.2:

" We remind that the Z-filter suppresses only the "anti-phase" 15N-1H RIDER, but not the "in-phase" one. However, the "in-phase" RIDER distortion of the COS component at reasonably short NTR is practically negligible, as our data demonstrate."

Comment: No pulse is used on 1H during the Z-filter. Will CW decoupling during this period help?

Reply: The Z-filter in fact allows for the decay of the SyIz coherence (the anti-phase term appearing after CP) due to spin-spin relaxation of nuclei S (15N or 13C). Strong proton CW decoupling during the Z-filter delay would make the filter worse since 15N(13C) T2 would become longer and one would have to increase the delay of the filter for suppressing the anti-phase coherence. CW decoupling under the rotary resonance conditions can decrease T2 and thus the delay can be shortened. This in principle can be done but we see no practical benefit from this complication of the experiment. Typical Z-filter delays of 50-100 ms anyway are much shorter than T1 and they bring no troubles with significant increase of experimental time and decrease of a signal, additional heating of samples, overload of amplifiers, power calibration, etc.

Comment: It is unclear what the mechanism for RIDER distortions in the in-phase signal for the CSA-CODEX experiment are (Fig 7, COS component). Is this attributable to imperfect decoupling during the CSA recoupling element?

Reply: This is correct. That was explained in the text of the paper, lines 288-291 of the original submission. To make it more clear we added one more sentence in this part of the text in the revised version:

" The Z-filter eliminates the anti-phase coherence (Eqs. 7 and 8), but it does not improve the efficiency of the proton decoupling during the de(re)phasing periods and thus the phase ïĄĘD remains non-zero."

Comment: The conclusions of this article are contrary to the authors' 2009 article on the same subject. This is not clearly bought out, I think it will be very important for the NMR community that this be pointed out clearly and discussed clearly in the manuscript, somewhere near Lines 405-415 (where this is partially done).

Reply: We agree, and add a sentence in the second paragraph of the section 4.2 clearly stating that our previous data on dipolar CODEX were misinterpreted:

" This means that the decay in the dipolar CODEX mixing time dependencies of backbone nitrogens in SH3 protein that we observed earlier (Krushelnitsky et al., 2009) is not due to molecular motions but due to the RIDER effect and that these data were misinterpreted."

Comment: The authors have previously stated (Reichert and Krushelnitsky 2018) that the in-phase component cannot be selected out in a CSA-CODEX experiment. Can this be clarified in light of Fig. 3?

Reply: This question appeared most probably again because of the unclear distinction between in-phase / anti-phase and COS / SIN signal decompositions. In our previous work we mentioned that for dipolar CODEX, COS and SIN components correspond to the in-phase and anti-phase coherences that appear under the influence of the X-H

dipolar interaction. Such the attribution is valid only for dipolar CODEX. (It should be also mentioned that this is valid only if the proton flip-flop time is much longer than NTR and if we neglect or suppress the anti-phase coherence that appears after CP. We did not mention this in the previous publication and now we correct ourselves.) For the case of CSA CODEX with an additional dipolar interaction this attribution is not valid anymore. Yes, we can of course measure COS and SIN components separately, but both of them would contain in-phase and anti-phase contributions. All this is considered in detail using the product operator formalism in the section 2.

Comment: L 215: "...instability of the MAS controller". How much is this instability, and presumably the rotor synchronization of the pulse sequence by monitoring the output of the mas controller should take care of this?

Reply: Usually, MAS controllers ensure the stability of about +/- 2-4 Hz. The MAS controller that was available for the experiments with BOC-Gly had unfortunately worse stability, +/- 5-10 Hz, sometimes MAS could spontaneously increase or decrease from the set value even on 20-50 Hz. For this reason, we had to repeat some of the measurements. The experiments were of course conducted in the rotor-synchronized mode, but the rotor synchronization ensures only the phase coherence at start points of the dephasing and rephasing periods. However, the delays between the recoupling pulses during the de(re)phasing periods were fixed. Thus, if the MAS rate deviates from the set value, then the recoupling efficiency during the de(re)phasing periods is somewhat different. This problem can be resolved if the software could read the actual MAS rate and calculate the delays between the recoupling pulses before each scan. Then the MAS rate instability of even few hundred Hz would not matter at all. Unfortunately, such an option was not available to us.

Comment: L 215-220: Does the five-point filter mean that each point shown in the figures is a average of five points, 2 before and 2 after the plotted point? Is this used only for visualisation or for fitting as well?

Reply: Yes, this is correct. We did not fit (or anyhow analyze quantitatively) the data presented in Figs. 5,7 and 8 at all.

――――――――――――――――――――